# Metabolic Parameters in Patients with Suspected Reactive Hypoglycemia

**DOI:** 10.3390/jpm11040276

**Published:** 2021-04-07

**Authors:** Marianna Hall, Magdalena Walicka, Mariusz Panczyk, Iwona Traczyk

**Affiliations:** 1Department of Human Nutrition, Faculty of Health Sciences, Medical University of Warsaw, 01-445 Warsaw, Poland; iwona.traczyk@wum.edu.pl; 2Department of Internal Diseases, Endocrinology and Diabetology, Central Clinical Hospital of the Ministry of Internal Affairs and Administration in Warsaw, 02-507 Warsaw, Poland; magdalena.walicka@cskmswia.pl; 3Department of Human Epigenetics, Mossakowski Medical Research Institute Polish Academy of Sciences, 02-106 Warsaw, Poland; 4Department of Education and Research in Health Sciences, Faculty of Health Sciences, Medical University of Warsaw, 02-091 Warsaw, Poland; mariusz.panczyk@wum.edu.pl

**Keywords:** reactive hypoglycemia, hyperinsulinemia, insulin resistance

## Abstract

Background: It remains unclear whether reactive hypoglycemia (RH) is a disorder caused by improper insulin secretion, result of eating habits that are not nutritionally balanced or whether it is a psychosomatic disorder. The aim of this study was to investigate metabolic parameters in patients admitted to the hospital with suspected RH. Methods: The study group (SG) included non-diabetic individuals with symptoms consistent with RH. The control group (CG) included individuals without hypoglycemic symptoms and any documented medical history of metabolic disorders. In both groups the following investigations were performed: fasting glucose and insulin levels, Homeostatic Model Assessment for Insulin Resistance (HOMA-IR), 75 g five-hour Oral Glucose Tolerance Test (OGTT) with an assessment of glucose and insulin and lipid profile evaluation. Additionally, Mixed Meal Tolerance Test (MMTT) was performed in SG. Results from OGTT and MMTT were analyzed in line with the non-standardized RH diagnostic criteria. Results: Forty subjects have been enrolled into SG. Twelve (30%) of those patients had hypoglycemic symptoms and glucose level ≤55 mg/dL during five-hour OGTT and have been diagnosed with RH. Ten (25%) subjects manifested hypoglycemic like symptoms without significant glucose decline. Patients with diagnosed RH had statistically significantly lower mean glucose at first (92.1 ± 37.9 mg/dL vs. 126.4 ± 32.5 mg/dL; LSD test: *p* < 0.001) and second (65.6 ± 19.3 mg/dL vs. 92.6 ± 19.3 mg/dL; LSD test: *p* < 0.001) hour of OGTT and insulin value (22.7 ± 10.9 lU/mL vs. 43.4 ± 35.0 lU/mL; LSD test: *p* < 0.001) at second hour of OGTT compared to the patients who did not meet the criteria of RH. Seventeen (43%) subjects from SG reported symptoms suggesting hypoglycemia during MMTT but none of them had glucose value lower than ≤55 mg/dL (68.7 ± 4.7 mg/dL). From the entire lipid profile, only mean total cholesterol value was significantly higher (*p* = 0.024) in SG in comparison with CG but did not exceed standard reference range. Conclusions: No metabolic disturbances have been observed in patients with diagnosed reactive hypoglycemia. Hyperinsulinemia has not been associated with glycemic declines in patients with this condition. Occurrence of pseudohypoglicemic symptoms and lower glucose value was more common after ingestion of glucose itself rather than after ingestion of a balanced meal. This could suggest an important role that nutritionally balanced diet may play in maintaining correct glucose and insulin levels in the postprandial period.

## 1. Introduction

The impairments of carbohydrate metabolism such as insulin resistance (IR), hyperinsulinemia, prediabetes and type 2 diabetes are increasingly becoming a serious, widespread health problem occurring in all social groups [1]. Moreover, reactive hypoglycemia, which refers to hypoglycemic symptoms and low blood glucose level that occurs in non-diabetic patients within 2–5 h after a meal [2,3] may be related to the above-mentioned conditions [4]. It has been observed in recent years that an increased number of individuals with symptoms suggesting hypoglycemia is being referred to the medical practitioners and/or dietitians [5,6].

Pathomechanism of hypoglycemia in non-diabetic subjects has not been widely studied, hence it is not yet fully understood. Due to this fact, there are no specific biochemical guidelines for the diagnosis. The threshold of glycemia, which defines hypoglycemia in person without diabetes, has not been clearly established; the values found in the literature vary from 40 mg/dL to 70 mg/dL [2,5,7,8]. Currently, in order to diagnose hypoglycemia, all the elements of Whipple triad (symptoms typical for hypoglycemia, confirmation of low blood glucose levels, resolving symptoms after carbohydrate intake) are required to be present [3]. However, these criteria raise many questions over the diagnostic tests which should be performed and specific threshold values for such biochemical tests in the relation to RH. The ability to measure the plasma glucose level at the accurate time of hypoglycemic postprandial symptoms is rare. It should be acknowledged that by the time the blood sampling is implemented the counter-regulatory hormones may influence plasma glucose concentration and the results will not be consistent with the submitted symptoms [9]. It is also challenging to replicate an environment, in which the patient presents the impairment in blood glucose concentration, in the ambulatory care settings. It should be emphasized that sole presence of hypoglycemic like symptoms without the actual confirmation of a significant glycemic decrease in the biochemical test, as well as low glucose levels without any coexisting symptoms, are not sufficient for reaching the final diagnosis [7].

Despite some controversies [2,7,10] the most widely implemented test for RH diagnosis is still OGTT [11]. Alternatively, the Mixed Meal Tolerance Test (MMTT) can also be performed to evaluate glucose decline and coexisting hypoglycemic like symptoms. Compared to OGTT when the patient ingests only glucose, MMTT contains protein, carbohydrates, and fat. This seems to imitate a typical meal in the patients’ daily diet better [12]. However, the definition of MMTT remains unclear; there are no specific guidelines according to meal composition and measurement that should be conducted during postprandial period. It should be highlighted that there is no gold standard for duration of these tests (neither OGGT nor MMTT) and subsequent interpretation of glucose and insulin values according to RH.

It is also difficult to clearly define RH. RH diagnosis requires exclusion of other conditions that may have impact on hypoglycemic episodes (e.g., previous gastrointestinal surgery, peptic ulcer disease, hormonal disorders, and alcoholism). The main drawback in RH diagnostic process is lack of standardized biochemical test and lack of glucose cut off value, which should be implemented. Currently, there are no published papers exploring this specific issue of adequate diagnostic measurements relating to RH. Additionally, a wide range of self-reported symptoms could be misleading and clinical presentation may vary in patients. It is unknown why some patients experience symptoms suggesting hypoglycemia even though blood glucose level during the measurement is correct. It is also challenging to determine why some patients have lower blood glucose levels during OGTT and do not manifest any symptoms. Potentially, an increased insulin sensitivity or over secretion of insulin (hypierinsulinemia) should be considered during the diagnosis but there is still not enough evidence to support that approach.

Decreased levels of blood glucose observed in patients with RH may suggest an early onset of glucose metabolism impairment, which may be associated with increased risk of developing diabetes [13]. It could also be viewed as a metabolic disorder but, similarly, there are not enough studies to support this hypothesis [14]. Moreover, there are no studies which have investigated possible lipid abnormalities in the patients with RH. However, in patients with metabolic disorders including increased insulin resistance, direct or indirect effects on lipid metabolism are observed. This may result in increased postprandial hyperlipidemia and increased triglyceride (TG) synthesis and reduced very low-density lipoproteins (VLDL) catabolism and decreased high-density lipoprotein cholesterol (HDL-C) concentration. Additionally, there is an increased risk of steatohepatitis as a result of insulin resistance in adipose tissue and increased fatty acid synthesis from glucose [15]. It is worth mentioning that these issues also affect slim (within normal BMI range) individuals who are commonly not associated with patients with metabolic disorders [16]. Therefore, assessment of glucose, insulin, and lipid profile may be crucial in patients with suspected reactive hypoglycemia. Taking into consideration all the above-mentioned issues, the aim of the study was to determine whether performed measurements of metabolic parameters have showed any difference in the patients with suspected reactive hypoglycemia compared to individuals without any metabolic disorders.

## 2. Materials and Methods

### 2.1. Participants

Initially, the study enrolled ninety patients admitted to the Department of Internal Medicine, Endocrinology and Diabetology of the Central Clinical Hospital of the Ministry of Internal Affairs and Administration in Warsaw between 2019 and 2020. Each patient who participated in the study has been hospitalized for three days. Participants were divided into two main groups—one study and one control group.

The selection criteria for SG were an occurrence of hypoglycemic like symptoms (hands tremor, increased sweating, palpitations, hunger, disorientation, impaired visual acuity, speech difficulty, confusion, impaired concentration and memory, incoordination, or fainting) which appeared in the postprandial periods. The age range for the participants was between 18 and 75 years old.

The exclusion criteria for SG and CG were the following: Type 1 diabetes (T1D), Maturity Onset Diabetes of the Young (MODY), Latent Autoimmune Diabetes in Adults (LADA), type 2 diabetes (T2D); prediabetes, liver, heart or kidney failure; diagnosed pancreatic tumors and other oncological diseases; sepsis; unrivalled endocrine disorders; condition after stomach or bowel resection; active stomach ulcer disease, pregnancy or menopause; treatment with drugs that may cause hypoglycemia, alcohol abuse.

CG participants were comparable with the SG individuals, in terms of gender, race, age and weight. After obtaining written informed consent, all subjects underwent biochemical examinations. Commission for Ethics and Supervision of Human and Animal Research at Central Clinical Hospital of the Ministry of Internal Affairs and Administration in Warsaw approved the study.

### 2.2. Anthropometric and Laboratory Measurements

The study data included anthropometric and biochemical measurements of seventy-five participants. Information relating to their medical condition, comorbidities and pharmacotherapy was collected during the admission to the hospital. Weight was evaluated with a standard scale and height was measured via a wall stadiometer. Body Mass Index (BMI) was evaluated based on the World Health Organization (WHO) recommendations: underweight (<18.5 kg/m^2^); normal range (18.5–24.9 kg/m^2^); overweight (25.0–29.9 kg/m^2^); obese (≥30 kg/m^2^) [17].

In the morning, of the first day of the hospital admission, fasting glucose, fasting insulin, OGTT and lipid profile measurements were performed. Lipid profile included total cholesterol (TC), high-density lipoprotein cholesterol (HDL-C), low-density lipoprotein cholesterol (LDL-C) and triglycerides (TG) measurements. TG/HDL-C ratio was evaluated for indication of atherogenic lipid profile.

The normal range of related indicators: fasting blood glucose (70–99 mg/dL), fasting insulin (2.6–24.9 uIU/mL), TC (<190 mg/dL), HDL-C (>40 mg/dL for men; >45 mg/dL for women), LDL-C (<115 mg/dL), TG (<150 mg/dL), and TG/HDL (≤3).

Before their admission to the hospital, all patients have been asked to carry on with their normal diet without any carbohydrate restriction, for at least three days prior to participating in the study. Patients were instructed to rest, get a full night’s sleep and not to undertake increased physical activity one day before the test was due. To participate in the study, patients had to be in the fasting state for at least eight hours. The procedure of OGTT required administering orally seventy-five grams of glucose. The blood samples were drawn to test glucose and insulin at the baseline (before glucose intake) and every sixty minutes for a period of five hours after ingestion of glucose. During the onset of symptoms attributed to hypoglycemia, patients were asked to mark these incidents on a prepared form.

The cut off glucose value for hypoglycemia during OGTT was set on ≤55 mg/dL [18].

Hyperinsulinemia was defined as an approximately tenfold increase of insulin level during five-hour OGTT [19].

RH was confirmed if the patient manifested hypoglycemic symptoms and had glucose level ≤55 mg/dL during OGTT.

Based on fasting glucose and fasting insulin levels Homeostasis Model Assessment of Insulin Resistance (HOMA-IR) was calculated to evaluate hepatic insulin sensitivity. HOMA-IR = fasting blood glucose (mmol/L) × fasting insulin (μIU/mL)/22.5 [20].

The cut off value for insulin resistance was HOMA-IR ≥ 2 [21].

On the following day, MMTT was performed only for the patients from SG. Patients consumed a non-liquid meal containing 60% of carbohydrate, 25% of fat and 15% of protein prepared by the dietician. The meal included three slices (75 g) of white bread, one tablespoon (10 g) of butter, three tablespoons (75 g) of semi-fat white cheese and two tablespoons (50 g) of jam. After the ingestion, patients were observed during a period of five hours and in case of occurrence of any symptoms suggesting hypoglycemia the blood sample was taken.

### 2.3. Final Characteristics of Participants

Fifteen subjects (five from SG and ten from CG) out of the initial group of ninety participants were excluded from the further study due to impairment of glucose metabolism—impaired fasting glycemia (IFG) or impaired glucose tolerance (IGT) revealed following OGTT, performed during the first day of testing. Their data has not been used in the analysis. For the purpose of this study data obtained from seventy-five patients (fifty-eight women and seventeen men) have been evaluated.

SG included forty subjects (thirty-three women and seven men) who reported symptoms attributed to hypoglycemia. Based on OGTT results, the study group was divided into two subgroups: patients with diagnosed RH (n = 12; nine women, three men) and patients without RH diagnosis (n = 28; twenty-four women, four men).

CG consisted of thirty-five asymptomatic individuals (twenty-five women, ten men), without any diagnosed metabolic disorders who were hospitalized for reasons unrelated to metabolic disorders and/or RH. This group was also hospitalized for three days as a standard hospital procedure.

The diagram provided in Figure 1 summarizes the protocol for selecting patients in the study, the performed procedures, and the subgrouping determined by the results of taken measurements.

### 2.4. Statistics

Anthropometric and biochemical parameters were presented using descriptive statistics (mean and standard deviation). Categorical variables were presented as numbers and percentages. The continuous variables were compared between groups using Student-*t* test or ANOVA with Fisher’s Least Significant Difference (LSD) post hoc test, depending on number of levels of a factor. The effect size for the observed difference between means was estimated with the help of Cohen’s d coefficient, whereby 0.2 is considered a ‘small’ effect size, 0.5 represents a ‘medium’ effect size and 0.8—a ‘large’ effect size. The chi-square (χ^2^) test was used to compare groups if there were categorical variables. The effect size was performed using the odds ratio with a 95% confidence interval. A receiver operating characteristic curve (ROC curve) as a graphical plot that illustrates the diagnostic ability of a binary classifier RH was used. The discriminant sensitivity and specificity were estimated by the area under the curve (AUC). A *p*-value of less than 0.05 was considered to indicate statistical significance. All analyses were performed with STATISTICA version 13.3 (TIBCO Software Inc., Palo Alto, CA, USA).

## 3. Results

### 3.1. Characteristics of the Studied Groups

The glucose value obtained during five-hour OGTT excluded eight patients due to impaired fasting glycemia (≥100 mg/dL) and seven patients due to impaired glucose tolerance (≥140 mg/dL). Overall, seventy-five individuals out of ninety have been enrolled in the study and their data has been evaluated. There was no significant difference in age (37.0 ± 9.9 vs. 33.8 ± 9.5; *p* = 0.162) and BMI (23.7 ± 3.0 kg/m^2^ vs. 24.9 ± 4.9 kg/m^2^; *p* = 0.198) between patients in SG and CG.

### 3.2. Fasting Glucose Concentration

There was no significant difference (*p* = 0.251) in mean fasting glucose concentration in SG and CG (85.0 ± 6.6 mg/dL vs. 86.6 ± 5.2 mg/dL). As well, there was no significant difference (*p* = 0.057) in mean fasting glucose concentration between patients with established diagnosis of reactive hypoglycemia (82.0 ± 6.2 mg/dL) and those without confirmation of the diagnosis (86.3 ± 6.4 mg/dL). In the first subgroup, only one subjects had a lower fasting glucose level than 70 mg/dL (67 mg/dL).

### 3.3. Fasting Insulin Concentration

There was no significant difference (*p* = 0.106) in mean fasting insulin concentration in the SG (7.9 ± 3.5 lU/mL) compared to CG (9.5 ± 5.8 lU/mL). As well, there was no significant difference (*p* = 0.056) in fasting insulin concentration between patients with RH and those without confirmation of the diagnosis (6.1 ± 2.5 lU/mL vs. 8.4 ± 3.6 lU/mL).

### 3.4. Insulin Resistance

Insulin resistance (HOMA-IR ≥ 2) was found in fifteen (38%) subjects in SG and fifteen (43%) subjects in CG. There was no significant difference (*p* = 0.081) in HOMA-IR compared SG and CG (1.7 ± 0.8 vs. 2.1 ± 1.4). HOMA-IR was statistically significantly lower (*p* = 0.029) in patients with RH diagnosis (1.2 ± 0.5) compared to those who did not met HR diagnosis (1.8 ± 0.8). Only one patient with confirmed RH had insulin resistance.

### 3.5. Oral Glucose Tolerance Test

Hypoglycemia has been found in twelve (30%) subjects from SG. Those patients have also manifested hypoglycemic symptoms during OGTT, therefore in this group RH has been diagnosed. Ten (25%) patients from SG had pseudohypoglycemic symptoms but without glucose decline during the test, and eight (23%) subjects from CG had biochemical hypoglycemia but without clinical manifestation. In SG, two patients had hypoglycemia in the first hour (in range 48–50 mg/dL), four in second hour (in range 43–53 mg/dL), six in third hour (in range 43–52 mg/dL) during OGTT. None of the patients from SG had hypoglycemia during fourth and fifth hour of OGTT. The patients from CG who have had glucose decline had hypoglycemia only at third hour (in range 36–55 mg/dL) during OGTT.

The subgroup with confirmed HR has statistically significantly lower mean glucose values during OGTT (*p* < 0.001), especially at the first and second hour, compared to those without documented HR and the control group (Figure 2). When analyzing mean glucose value in the SG subgroups during OGTT, patients with confirmed RH had statistically significantly lower mean glucose at first (92.1 ± 37.9 mg/dL vs. 126.4 ± 32.5 mg/dL; LSD test: *p* < 0.001) and second (65.6 ± 19.3 mg/dL vs. 92.6 ± 19.3 mg/dL; LSD test: *p* < 0.001) hour compared to patients without diagnosed RH.

There were no statistically significant differences between mean insulin values between all three groups (*p* = 0.234) during five-hour OGTT (Figure 3) with an exclusion of second hour of measurements. Analyzing mean insulin value in the SG subgroups during OGTT, patients with diagnosed RH had statistically significantly (LSD test: *p* < 0.001) lower mean insulin levels at second hour of OGTT compared to patients who did not meet the criteria of RH (22.7 ± 10.9 μIU/mL vs. 43.4 ± 35.0 μIU/mL). Hyperinsulinemia has been found in sixteen (40%) subjects from SG (six with confirmed RH and ten without RH diagnosis) and eleven (31%) subjects in CG. There was no significant difference (*p* = 0.440) in hyperinsulinemia occurrence in SG and CG, as well as in patients with confirmed RH and those who did not meet the diagnostic criteria of RH (*p* = 0.398).

### 3.6. Specificity and Sensitivity of OGTT

The AUC was 0.261 (95%CI = 0.097–0.424; *p* = 0.004) for the first hour and 0.163 (95%CI = 0.025–0.301; *p* < 0.001) for the second hour of five-hour OGTT. The third, fourth and fifth hour of OGTT has no discrimination value (Figure 4).

For the challenge test, the cut off was determined to be 101 mg/dL at first hour of OGTT, with a sensitivity of 33% [95%CI: 9.9–65.1%] and a specificity of 33% [95%CI: 21.9–46.3%]. For the second hour of OGGT, the glucose threshold was 102 mg/dL with a sensitivity 8% [95%CI: 0.21–38.5%] and specificity of 57% [95%CI: 44.1–69.5%].

### 3.7. Mixed Meal Tolerance Test

The Mixed Meal Tolerance Test was performed only in SG, as additional tests for those patients with suspected reactive hypoglycemia were performed. Only seventeen (42%) subjects—nine with RH diagnosis and eight from the subgroup without RH confirmation had manifested symptoms attributed to hypoglycemia during the test and had their blood drawn to determine glucose level. The mean glucose concentration in SG was 68.7 ± 4.7 mg/dL. Four patients had unspecific symptoms at second hour, six at third hour and seven at fourth hour after ingestion of the meal. There was no significant difference (*p* = 0.206) in glucose value during MMTT in patients with confirmed RH (67.3 ± 4.11 mg/dL) compared to patients without confirmation of RH diagnosis (70.3 ± 5.0 mg/dL).

### 3.8. Specificity and Sensitivity of MMTT

The test was evaluated for sensitivity and specificity, but the small sample size was a limiting factor therefore the test produced a sensitivity of 75% [95%CI: 42.8–94.5%] and specificity of 71% [95%CI: 51.3–86.8%].

### 3.9. Lipid Profile

Comparing lipid profile in SG and CG only mean TC (176.8 ± 22.3 mg/dL vs. 163.5 ± 27.7 mg/dL) was statistically significantly higher (*p* = 0.024) in the first group. Considering the subgroup of patients with and without a diagnosis of reactive hypoglycemia, there were no significant differences in lipid parameters. The differences between groups and subgroups are highlighted in Table 1.

## 4. Discussion

It should be highlighted that our research is one of the very few studies which analyzes metabolic parameters in patients with suspected RH. Our research has shown that only a small percentage of patients have low glucose level during OGTT. Previous studies which have analyzed RH occurrence based on OGTT were conducted in the heterogeneous population. Those investigations involved the following: patients in various age groups and BMI range [22], healthy subjects [12], pregnant women [23], and patients with a variety of medical conditions [24,25,26]. The results obtained in those studies are consistent with our results and confirmed relatively low frequency rate of hypoglycemia occurrence during OGTT. In addition, in our study OGTT has also showed low sensitivity and specificity for RH diagnosis. On this basis, it is reasonable to conclude that assessment of glucose declines by performing OGTT is not an adequate diagnostic tool for RH. This approach is also consistent with claims made by other researchers in earlier studies [7,27]. Furthermore, it should be pointed out that earlier studies in the field of RH had different glucose cut off value which could have led to significant different findings [2,4,6,23,28]. Raising or lowering the glycemic cut off value can affect the final diagnosis. Lack of standardized hypoglycemic cut off value in the postprandial state in non-diabetes subjects is, without a doubt, the most important limitation during the diagnosis. Until a specific glucose value is established, discrepancies in RH diagnosis will continue to be significant.

In our study, the occurrence of insulin resistance and hyperinsulinemia were similar in the study and control groups. The subgroup with confirmed RH had even lower insulin values during OGTT compared to those who did not meet the criteria of RH. This outcome is similar to the result of other studies which evaluated insulin values in patients with RH and no abnormalities in insulin response were noted [29,30]. Therefore, hyperinsulinemia and insulin resistance do not necessarily have to be responsible for the occurrence of hypoglycemia. However, there are studies with HR patients who had insulin resistance and significant impairment of insulin secretion [31,32]. Due to discrepancies in the findings of the studies and considering the lack of up-to-date evidence from other publications, further research focusing on insulin secretion and insulin sensitivity in RH patients is required. It is still debatable if the reduction of insulin sensitivity leads to hyperinsulinemia or whether the increased insulin level during fasting and postprandial conditions could cause decrease of cell’s metabolic response to insulin [33]. For this reason, it would seem reasonable to verify the cut off values for fasting insulin, insulin resistance, and hyperinsulinemia which may affect investigation of RH. It must be acknowledged, that according to some researchers the acceptable fasting insulin levels should be lower and kept in the range between 5 and 15 μIU/mL [34]. This indicates the need for more precise norm for fasting insulin level evaluation, especially in patients with normal body weight. Moreover, debate continues about more precise threshold for insulin resistance and hyperinsulinemia in subjects with normal body weight [20,21,35]. It should be noted that additional components such as body composition, sex, age, race and ethnicity may influence fasting and postprandial insulin values and insulin sensitivity [36]. Furthermore, the lack of a precise definition of hyperinsulinemia in the context of the OGTT measurements contributes to the discrepancies in the diagnostic rate. The arbitrary values which were adopted in our study (a tenfold increase of insulin) seem to be consistent with the physiological assessment of insulin secretion during OGTT [19].

It should be highlighted that in our study, 23% of patients from the CG had hypoglycemia during OGTT but did not manifest any hypoglycemic symptoms. As well as the well-established counter-regulatory mechanisms in the fasting and postprandial state which are required to maintain stable glucose concentration in the blood, it is necessary to consider individual factors leading to occurrence of hypoglycemic like symptoms [37]. There were some studies which confirmed biochemical hypoglycemia in the individuals with no hypoglycemic symptoms due to a decreased sympathetic response [3,9]. Our research also has shown that 55% subjects in SG manifested symptoms suggesting hypoglycemia during OGTT but only half of them had hypoglycemia in the biochemical test. Symptoms suggesting hypoglycemia in absence of glucose decline were also observed in different research, which might suggest that the onset of pseudohypoglycemic symptoms could be an individual indicator, unrelated to any metabolic disorder [38]. Therefore, if no abnormalities in insulin secretion are observed, the psychological link should be considered as a main aspect responsible for the occurrence of unspecific symptoms. Psychosomatic disorders, emotion dysregulation and sleep disturbance can all potentially lead to pseudohypoglycemic symptoms without any glucose decline [39]. It should also be mentioned that there is an association between general mental health, eating patterns and unspecific symptoms which occurs in the postprandial periods [40]. Moreover, improper eating patterns, low energy diet, excessive alcohol intake may all lead to sudden glucose fluctuation and increased sympathetic neural activity [28,41].

In our study, MMTT was performed as an additional diagnostic tool for RH diagnosis. None of the patients who manifested symptoms suggesting hypoglycemia had hypoglycemia during this test. This finding is comparable with another study which compared glucose value during OGTT and MMTT in patients with suspected RH [42]. Both studies have shown that during OGTT the nadir glucose was lower than during MMTT. However, it is crucial to point out that there are no further published studies that are suitable to use when comparing results of OGTT and MMTT in patients with suspected RH. Additionally, it is worth noting that half of the patients in our study who had symptoms attributed to hypoglycemia during MMTT were diagnosed with HR based on OGTT. It would seem that glucose fluctuation during OGTT and MMTT may be dissimilar due to differences in the consistency (liquid vs. solid) and the composition (simple sugar vs. carbohydrate, protein, fat) of the ingested source of glucose which may affect the postprandial outcome. Assuming the same cut-off glucose value for OGTT and MMTT may lead to false negative results in MMTT. It indicates the need of differential glucose cut off value for both tests. Considering that MMTT represent more accurately a typical meal consumed by the patient, rather than the liquid containing seventy-five grams of glucose, it is recommended that additional studies should be conducted to improve the detection of RH via MMTT.

In our study, no significant differences were found between lipid profile dysregulation and RH. It is important to emphasize that our lipid reference norms are the standard norms for people without increased cardiovascular risk. The lack of additional clinical trials in this area of RH also affects the ability to conduct reliable evaluations. However, it is important to evaluate lipid profile due to increasing problem of metabolic disorders in non-obese patients [43]. The incorrect TG/HDL ratio shows greater risk of occurrence of IR markers in normal-weight adults without type 2 diabetes [16]. Disturbed insulin secretion and impaired response of pancreatic β cells affect not only glucose fluctuations but also cholesterol metabolism which leads to TG elevation and HDL-C level lowering [44]. Moreover, hepatic insulin resistance plays significant role in progression of dyslipidemia which, in turn, leads to a greater risk of developing metabolic disorders [45]. In addition, patients who already have a metabolic disorder may have an accelerated progression of the disorder [46].

The findings of this study have to be considered while taking into account some limitations that are present. The first, being a lack of glucose and insulin measurements during the whole duration of MMTT. Patients were observed for five hours after they have consumed a prepared meal, but glucose level was only evaluated when symptoms suggesting hypoglycemia were reported. Glucose and insulin measurements were not taken each hour like in OGTT procedure. For this reason, the tests cannot be fully compared. The comparison refers only to glucose level at the exact moment of taken measurement during both tests. When considering the differences in glucose levels obtained in OGTT and MMTT, it would appear that insulin levels should also be compared in those tests. The second limitation is the inconsistency in self-reported symptoms by the patients during OGTT and MMTT. Perceived symptoms are a subjective, therefore they may have been overlooked or exaggerated by the patients. Patients who were screened for RH were significantly more likely to experience symptoms of hypoglycemia despite no decline in glucose level. The individual’s perception of symptoms attributed to hypoglycemia, as well as the physiological attitude that led to somehow increased expectation of hypoglycemic episode, should also be considered.

It is recommended that further research is undertaken in the area of reactive hypoglycemia. Considerably more work is required to determine additional diagnostic procedures which could, in turn, improve the explanation of sudden occurrence of symptoms attributed to hypoglycemia in non-diabetic patients with and without hypoglycemia in the biochemical test. Lastly, the most important limitation lies in the fact that cut-off glucose value which should clearly evaluate hypoglycemia in postprandial period is not yet standardized. Until a standardized glucose cut off value is established, discrepancies in RH results in various studies will continue to be significant.

## 5. Conclusions

Based on the results obtained in our study, which have also been compared with data from earlier studies, we believe that reactive hypoglycemia is not associated with any metabolic disturbance. This would suggest that RH is related to individual factors that contribute to the severity of symptoms suggesting hypoglycemia. Moreover, hyperinsulinemia is not responsible for glycemic declines or occurrence of symptoms suggesting hypoglycemia in patients with confirmed diagnosis. These findings were validated by the reported changes in glycemia and insulinemia during OGTT and MMTT. It appears that the source of glucose may influence postprandial glycemic fluctuations and the onset of symptoms suggesting hypoglycemia. Therefore, dietary modifications may be an effective factor in reducing symptoms attributed to hypoglycemia.

## Figures and Tables

**Figure 1 jpm-11-00276-f001:**
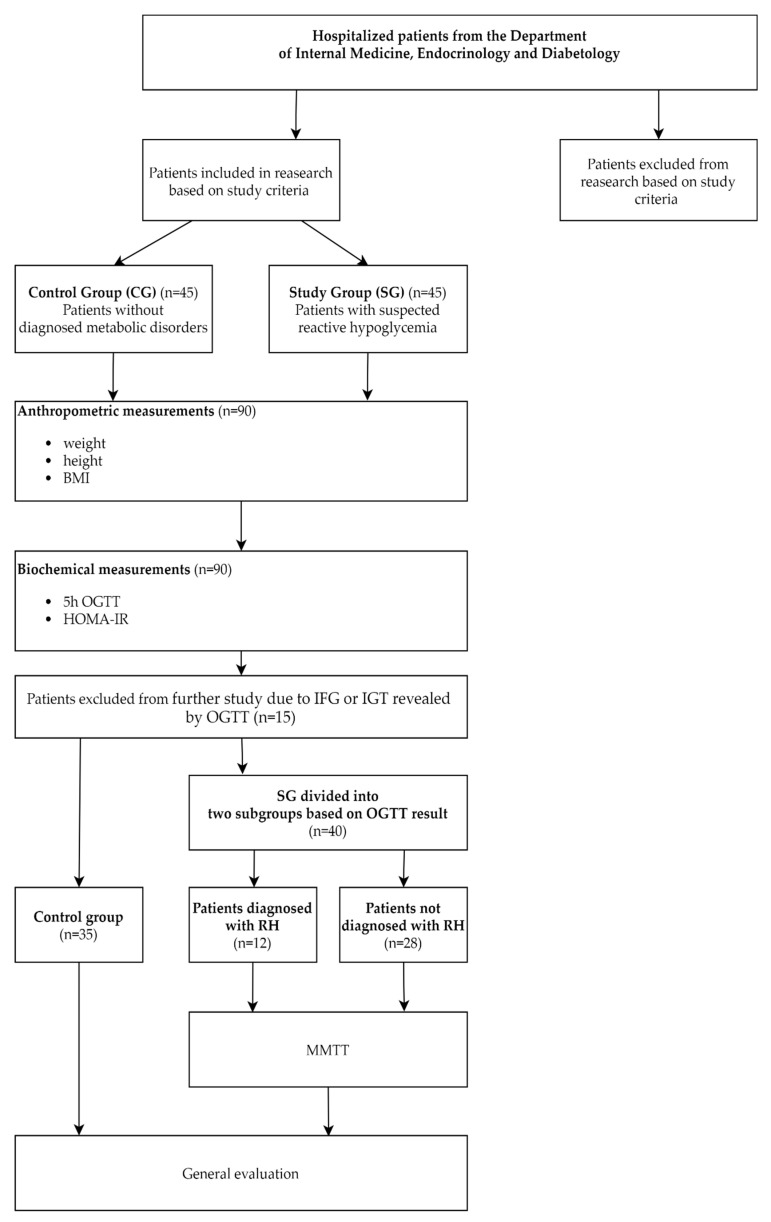
Performed procedures. BMI—Body Mass Index; OGTT—Oral Glucose Tolerance Test; HOMA-IR—Homeostatic Model Assessment for Insulin Resistance; IFG—Impaired Fasting Glycemia; IGT—Impaired Glucose Tolerance; RH—Reactive Hypoglycemia; MMTT—Mixed Meal Tolerance Test.

**Figure 2 jpm-11-00276-f002:**
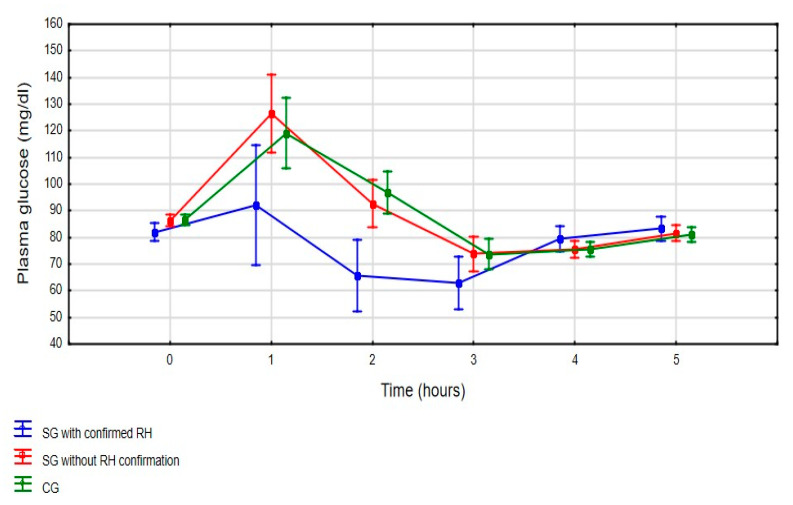
Mean glucose value during 5-h OGTT (three-way ANOVA: F_(10, 360)_ = 3.598, *p* < 0.001) [mean glucose value with a 95% confidence interval]. SG—Study Group; CG—Control Group; RH—Reactive Hypoglycemia.

**Figure 3 jpm-11-00276-f003:**
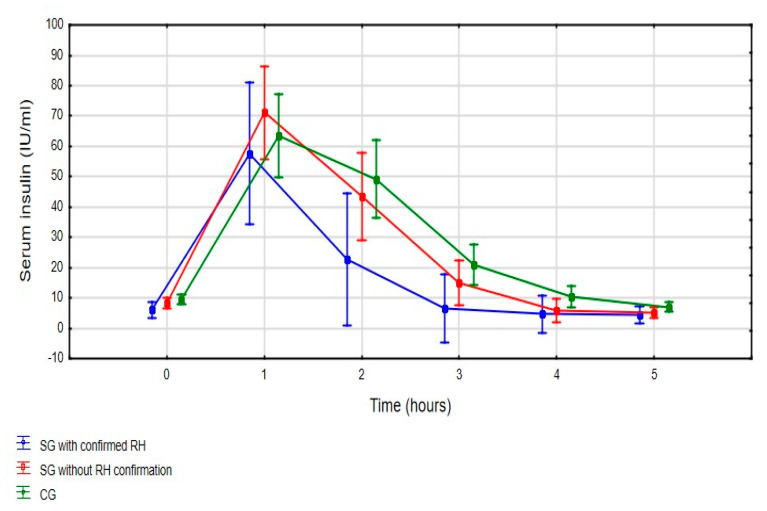
Mean insulin values during 5-h OGTT (three-way ANOVA: F_(10, 360)_ = 1.290, *p* = 0.234) [mean insulin value with a 95% confidence interval]. SG—Study Group; CG—Control Group; RH—Reactive Hypoglycemia.

**Figure 4 jpm-11-00276-f004:**
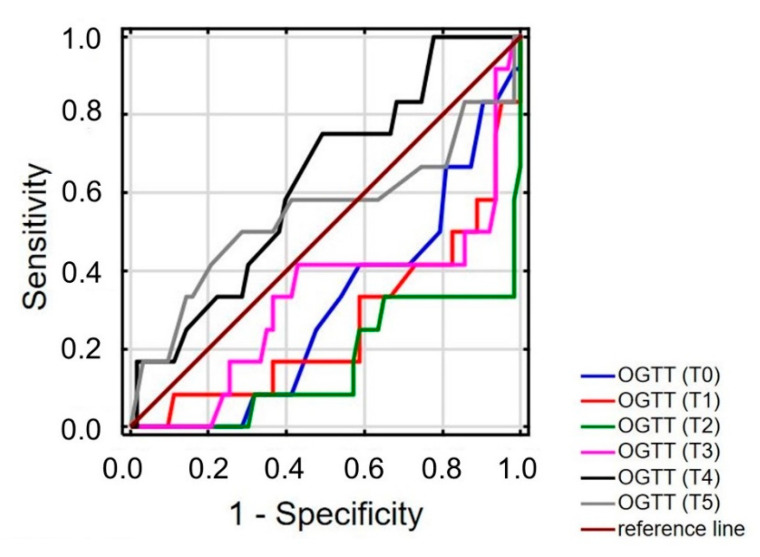
ROC graph for patients with reactive hypoglycemia. OGTT—Oral Glucose Tolerance Test; T0—measurement taken in fasting state; T1—first hour of measurement, T2—second hour of measurement; T3—third hour of measurement; T4—fourth hour of measurement; T5—fifth hour of measurement.

**Table 1 jpm-11-00276-t001:** Lipid profile.

Variables	Study Group (*n* = 40)	Control Group (*n* = 35)	*p* Value *	Non-Reactive Hypoglycemia (*n* = 28)	Reactive Hypoglycemia (*n* = 12)	*p* Value *
**TC**	176.8 ± 22.3	163.5 ± 27.7	0.024	176.0 ± 23.3	178.5 ± 20.7	0.750
**LDL-C**	97.5 ± 20.3	88.0 ± 23.8	0.068	99.2 ± 22.0	93.4 ± 15.7	0.414
**HDL-C**	60.1 ± 15.9	58.1 ± 14.0	0.560	57.5 ± 15.8	66.3 ± 14.8	0.106
**TG**	89.3 ± 36.6	85.1 ± 41.2	0.649	92.9 ± 34.0	80.8 ± 40.2	0.342
**TG/HDL**	1.7 ± 1.0	1.6 ± 1.0	0.776	1.8 ± 1.0	1.4 ± 1.1	0.245

TC—total cholesterol (mg/dL); LDL-C—low-density lipoprotein cholesterol (mg/dL); HDL-C—high-density lipoprotein cholesterol (mg/dL); TG—triglycerides (mg/dL); * Student’s *t*-test.

## Data Availability

The data presented in this study are openly available on “Zenodo” at https://doi.org/110.5281/zenodo.4666896, accessed on 1 April 2021.

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
