# Peer review of "Metabolic Parameters in Patients with Suspected Reactive Hypoglycemia"

_jpm, 2021, doi:10.3390/jpm11040276_

Round 1

Reviewer 1 Report

The manuscript aimed to investigate metabolic changes in patients with idiopathic reactive hypoglycemia. The subject of the mechanism involved in this type of hypoglycemia is of great interest for clinicians as currently is still unknown. Previously it was hypothesized that insulin resistance with higher fasting/preprandial insulin levels or an exaggerated and rapid increase of insulin release. The research submitted by Hall et al showed that patients with reactive hypoglycemia had fasting insulin concentration levels similar to the ones of patients without reactive hypoglycemia and furthermore HOMA-IR and insulin levels during the OGTT were significantly lower in patients with reactive hypoglycemia. Thus, by these findings the authors showed that previous hypotheses on the involvement of insulin resistance and increased insulin secretion as mechanisms of reactive hypoglycemia are not supported by evidence.

The research was carefully designed, the manuscript is very well written, clear and easy to read. The conclusions are consistent with the results presented and they all address the main question of the research.

Author Response

Dear Reviewer,

Thank you for your letter and taking the time to revise our paper “Metabolic parameters in hospitalized patients with suspected reactive hypoglycemia” on the Journal of Personalized Medicine.

We greatly appreciate your review, as well as the positive feedback in relation the submitted  abstract and the contents of our paper.

We have now implemented some further changes which have been suggested by another reviewer. These changes have been approved by all the authors of the paper and have been highlighted in the manuscript as requested. Herewith, the revised version.

You will note that we have added a paragraph highlighting that, so far, there has been very few publications of this natures, which can support a hypothesis relating to occurrence of reactive hypoglycemia (line 96).

We have also explained the reason why the number of subjects/patients initially considered for the study, and those who in fact participated, has reduced from 95 to 70 (line 123). This change is also reflected in the Figure 1.

We have now ensured it is clear to the reader that the data obtained from those subjects who did not continue with the trial were not included in the evaluation of the results (line 127; 211).

You will note that we have also added information relating to the gender distribution of the  participants who were hospitalized during our study (line 128; 130; 132; 133; 134).  

The reference list has been slightly reduced, however not all publications older than 20 years have been removed. This is due to the fact that there is still significant lack of more recent, valuable data which addresses to reactive hypoglycemia.

Lastly, we have now added the reference number of the Ethic Committee approval, which was missing in the earlier draft. We apologize for not including it previously; it was an oversight on my part.

We can confirm our article has been now reviewed by a native speaker  and necessary corrections have been made.

We trust that the modifications mentioned above are acceptable, but should you wish to add any further comments, we will be grateful for your continuous guidance.

We hope the revised manuscript meets the Journal of Personalized Medicine criteria and requirements and is one step closer to being published.

Thank you for your continued support and interest in our research.

Sincerely,

Marianna Hall

Reviewer 2 Report

Thank you for the opportunity to review the manuscript “Metabolic Parameters in Hospitalized Patients with Suspected Reactive Hypoglycemia” by Marianna Hall et al. It is an interesting issue met in the patients of all age and therefore studies trying to explain the possible metabolic background of reactive hypoglycemia (RH) are valuable.
The authors studied some of the metabolic parameters in patients with RH and compared them to the control group.
However, there are some very crucial issues that in my opinion should be improved before deciding whether the manuscript is eligible for the publication.
Major comments:
1. The hypotheses should be presented in a more clear way. In the Introduction the reader does not receive the proper information regarding the possible influence of lipid profile or hyperinsulinaemia on RH. Therefore, the aim of the research is not properly state. It is explained late in the manuscript, in the Discussion.
2. There are major wrong descriptions of the groups:
- Comparing the text and the Figure 1: in the Fig. 1 we have one CG 45 pts, than CG 35 pts and this is very misleading.
- Moreover, the anthropometric and biochemical data and analysis regards 90 or 75 pts? This is crucial for the conclusions.
- In the title we have the information that the study was conducted in “hospitalized patients”. It rather suggests that the patients were hospitalized due to other conditions. As far as I understand, it is just the research was performed in the hospital settings, which is a very different situation.
- The reasons for patients exclusion should be stated in the Materials and Methods not only in the Results.
- There is no data on the sex of the patients!
3. The authors should not describe “hypoglycemia symptoms” if the background of the symptoms is not hypoglycemia. They are rather “pseudohypoglycemic symptoms” or “symptoms suggesting hypolycemia”.  
4. There is lack of ref. and/or comment on the norms for lipid profile, OGTT, HOMA-IR results. It is crucial for the lipid profile to interpret the results in view of possible cardiovascular risk or even the sex of the patients.
5. The results (paragraphs 3.3-3.5) would be more clear if the authors put them in a Table.
6. Not all conclusions are justified and supported by the results. The conclusions can be only formed with support of the presented results, which are lipid profile and insulin/HOMA-IR values in the OGTT and MMT tests. In the Conclusions there is no comment on the difference in TG levels and it’s possible importance.
7. The reference list is definitively too long and not fully up-to-date! There shouldn’t be more than 30-40 references and they should be written in English. The references should not be older than app. 15-20 years, unless there is an exceptional cause, which I cannot find in the manuscript. The ref. should be stated at the end of the sentence otherwise it is very hard to comprehend the text. In the Discussion there are several ref. put in one sentence which make it very unclear.
8. The text should be more concise and precise in general. The description of some of the procedures, for example BMI measurement, OGTT in such details is not necessary, as they are standard tests.
9. There is lack of number of Ethic Committee approval.
10. The English language needs major corrections in grammar and style.

Minor comments:
1. Unclear phrases: line 104 “tumors of pancreatic island”, line 274 “sensitivity and sensitivity”, line 370 “regulated glucose value”.
2. Unclear abb.: line 97, Fig. 1 “MSWiA”, Fig 2 and 3 “RH yes, RH no”.

Author Response

Dear Reviewer,

Thank you for your recent message. We appreciate your thorough review and detailed   comments relating to our article titled “Metabolic parameters in hospitalized patients with suspected reactive hypoglycemia” submitted to the Journal of Personalized Medicine. 

All of the reviewer’s suggestions have been immensely helpful. We have now reviewed our paper and implemented all of the necessary changes.

We have also taken this opportunity to respond to your comments here to ensure the concerns you have raised are addressed appropriately.

Any changes made have been approved by all of the authors but, once again, I have been appointed as the corresponding author. The changes have been highlighted in red, throughout the body of the paper, per your earlier request. Herewith the revised draft of our manuscript for your further consideration.

Comment 1: The hypotheses should be presented in a more clear way. In the Introduction the reader does not receive the proper information regarding the possible influence of lipid profile or hyperinsulinemia on RH. Therefore, the aim of the research is not properly state. It is explained late in the manuscript, in the Discussion.

Response 1: Thank you for this comment.  Since the process responsible for the occurrence of RH is still largely unknown and there is a very small number of studies/ data available relating to this topic we have added a paragraph that highlights this fact (line 96). 

Additionally, it is worth pointing out that there is a significant lack of studies that address the lipid disturbances which could conclusively support the hypothesis of a possible influence of lipid disturbance and RH occurrence. 

For this very reason, we have decided to perform an analysis of insulin secretion (evaluation of insulin resistance and hyperinsulinemia) as well as analysis of lipid values in order to investigate the relationship between disturbances in insulin and/or lipid values and the possibility of the occurrence of RH. 

Comment 2: There are major wrong descriptions of the groups:

Response 2: Thank you for pointing this out. We have reviewed and clarified the following points below.

  1. Comparing the text and the Figure 1: in the Fig. 1 we have one CG 45 pts, than CG 35 pts and this is very misleading.
  1. To clarify this point, we have slightly amended Fig. 1 as follows: we have changed “Patients excluded from research due to IFG and IGT” to “Patients excluded from further research due to IFG and IGT revealed by OGTT (n=15)”. 

We trust this explains better the reason for reduction in CG from 45 to 35, and in SG from 45 to 40, which took place in the course of the study. 

We have also added a sentence (line 124 ) in Materials and Methods (2.1. Participants) so it is consistent with the information within the body of the paper and in the Figure 1. 

We have kept the sentence in Results unchanged as it was already stating the correct information mentioned above (line 211).

  1. Moreover, the anthropometric and biochemical data and analysis regards 90 or 75 pts? This is crucial for the conclusions.
  1. In order to be more transparent when discussing the number of participants and the final data which has been analyzed we have added a sentence (line 127) which emphasizes that the results we obtained include only the data (n=75) from the participants who were not excluded during the first day of the testing.

The sentence in Materials and Methods (2.1. Participants) with reference to the number of participant remained without changes. 

Additionally,  a minor change has been done in section Results (3.1. Characteristics of the studied groups) so that it also highlights the number of participants included in the study (line 211).

  1. In the title we have the information that the study was conducted in “hospitalized patients”. It rather suggests that the patients were hospitalized due to other conditions. As far as I understand, it is just the research was performed in the hospital settings, which is a very different situation.
  1. You have raised an important point here. To clarify, all of the participants were hospitalized in the Department of Internal Medicine, Endocrinology and Diabetology of the Central Clinical Hospital of the Ministry of Internal Affairs and Administration in Warsaw.

Participants from SG were admitted to the hospital due to the symptoms indicating reactive hypoglycemia, hence the reason why they have been referred to the hospital, for potential reactive hypoglycemia diagnosis. 

Participants from CG were, however hospitalized due to other conditions, which were not related to metabolic disturbance or reactive hypoglycemia. This information was also added to the sentence describing CG (line 135).

In order to highlight the point that these patients were all hospitalized, we have added further information regarding their duration of hospitalization within the text (line 109).

  1. The reasons for patients exclusion should be stated in the Materials and Methods not only in the Results.
  1. Noted and agreed. We have now added the additional exclusion criteria in the Materials and Methods (line 126) . 

We have also emphasized that patients, who were initially included in the study had to be withdrawn from the further study due to incorrect results obtained during the OGTT, which was performed during their hospitalization.

  1. There is no data on the sex of the patients.
  1. Noted and agreed. We have, therefore included the data on the sex of the patients, in brackets. For consistency purposes, this data was stated in the description of the control group, the study group and the subgroups of the study group (line 128; 130; 132; 133; 134). 

Comment 3: The authors should not describe “hypoglycemia symptoms” if the background of the symptoms is not hypoglycemia. They are rather “pseudohypoglycemic symptoms” or “symptoms suggesting hypolycemia”.  

Response 3: Noted and agreed. We have incorporated your suggestion throughout the manuscript. The relevant amendments have been made to all sentences that contained "hypoglycemia symptoms" if the background of the symptoms was not associated with patients’ glucose decline. 

Comment 4: There is lack of ref. and/or comment on the norms for lipid profile, OGTT, HOMA-IR results. It is crucial for the lipid profile to interpret the results in view of possible cardiovascular risk or even the sex of the patients.

Response 4: Information about the norms for lipid profile was provided in section Materials and Methods (2.2. Anthropometric and laboratory measurements; line 156; 171; 172; 174; 176; 179). These are the values that are accepted (as the norms) by the hospital laboratory where the results were analyzed.

We have clarified this in the manuscript by separating reference to the norms into separate paragraphs.

Additionally, comments regarding cut off values for the glucose during OGTT, as well as for HOMA-IR and hyperinsulinemia values, are discussed in the Discussion section. This is highlighted as a factors that can lead to discrepancies in the final diagnosis.

We are aware that our lipid values do not take into account norms that are variable for patients with cardiovascular risk. However, all patients included in the study were young and with no previous cardiovascular history, therefore we have decided to apply norms that are generally accepted for the population. 

Comment 5: The  results (paragraphs 3.3-3.5) would be more clear if the authors put them in a Table.

Response 5: Thank you for this comment which we have considered.  However, we would prefer to keep the graphs, which in our opinion more clearly show the dynamic changes of glucose and insulin values throughout five-hour OGTT. We trust that the readers will see the visualization of glucose and insulin fluctuation during OGTT which is crucial. The figures that are statistically significant have been provided in the text.

Comment 6: Not all conclusions are justified and supported by the results. The conclusions can be only formed with support  of the presented results, which are lipid profile and insulin/HOMA-IR values in the OGTT and MMT tests. In the Conclusions there is no comment on the difference in TG levels and it’s possible importance.

Response 6: We note and agree with the comment regarding the consistency of the conclusion. We have decided to present it in a more precise way. Following the amendments the conclusions are now supported by the presented results, which were crucial for our study (line 425).

Comment 7: The reference list is definitively too long and not fully up-to-date! There shouldn’t be more than 30-40 references and they should be written in English. The references should not be older than app. 15-20 years, unless there is an exceptional cause, which I cannot find in the manuscript . The ref. should be stated at the end of the sentence otherwise it is very hard to comprehend the text. In the Discussion there are several ref. put in one sentence which make it very unclear.

Response 7: Noted and agreed. We have incorporated your suggestion throughout the manuscript. The reference list has been reduced. 

Not all publications older than 15-20 years have been removed due to the fact that there is a lack of more recent, valuable research on the subject. Considering that the phenomenon of RH remains unclear and the results obtained in those older studies are consistent with the results reported in our study,  we  have made a decision that they should be kept in the manuscript. Consequently, we also highlighted this in our publication (line 302; 369; 384).

Comment 8: The text should be more concise and precise in general. The description of some of the procedures, for example BMI measurement , OGTT in such details is not necessary, as they are standard tests.

Response 8: Thank you for this suggestion. Changes have been implemented in relation to BMI; the BMI formula have been removed.  

Considering that a two-hour OGTT is commonly performed, we wanted to describe the procedures that were conducted during the extended five-hour OGTT - especially considering the number of blood samplings. 

Comment 9: There  is lack of number of Ethic Committee approval.

Response 9: Noted. We have added the appropriate number of Ethic Committee approval, which was missing in the earlier draft. We apologize for not including it previously; it was an oversight on our part.

Comment 10: The English language needs major corrections in grammar and style.

Response 10: Noted. We can confirm our article has been now reviewed by a native speaker  and necessary corrections have been made.

Comment 11: Minor comments:

Response 10: Noted and agreed. We have modified those phrases.

Unclear phrases: 

  1. line 104 “tumors of pancreatic island” – changed - pancreatic tumors 
  2. line 274 “sensitivity and sensitivity”- changed - sensitivity and specificity
  3. line 370 “regulated glucose value” – changed - glucose cut off value

Comment 12. Unclear abb.: 

Response 12: Noted. We have modified those phrases.

  1. a) line 97, Fig. 1 “MSWiA ” – removed 
  2. b) Fig 2 and 3 “RH yes, RH no”. – changed into full explanation

We trust that these amendments are acceptable but should you feel more changes are necessary, do not hesitate to contact us. 

We hope the attached updated manuscript will match the expectations and standard of the Journal of Personalized Medicine.

Thank you for your continued assistance and interest in our research.

Sincerely,

Marianna Hall 

Round 2

Reviewer 2 Report

Dear Authors,

Thank you for the effort to improve the manuscript. I find the corrections significant. However, there are still some issues that should be considered before the submission.

Major comments:

  1. In my opinion the title should be modified and it should not include “hospitalized”. As explained by Authors only the CG was hospitalized per se (lines 135-136). SG is the clue of the research and these patients were hospitalized only to perform the study/perform planned tests.

- lines 109-110- there is again inaccuracy, whether the CG was also hospitalized for 3 days? It seems to be different from the information in the lines 135-136.

  1. Lines 96-100: thank you for adding the paragraph. It should contain ref. Moreover, I still think there should be mentioned hypothesised role of lipids in glucose/insulin metabolism (as in lines 387-390) in the introduction, as this is a crucial part of the research.
  2. I appreciate the changes made in the explaining the study groups. However, Material and Methods section is still mixed: lines 124-136 should go after line 179 as some of the patients were excluded after OGTT, if I understand correctly.
  3. Lines 171-173: lack ref. As hypoglycaemia and hyperinsulinaemia are the important possible results of the research the reader should be given the ref. for the values adapted in the study.
  4. I understand that 5-hr OGTT is not a standard OGTT. However, OGTT is a widely known procedure and in my opinion most of the lines 163-170 should be deleted, except the information that: 75g of glucose was used and glucose and insulin levels were measured at baseline and every 60 minutes for a period of five hours.
  5. I feel there is lack of possible conclusions from the presented results: lower insulin in RH group suggests adequate insulin response, lower glucose in OGTT in comparison to MMTT suggests the reason for modifying diet in the patients with RH. Therefore, Conclusions in Abstract and the text should be modified, not only revealing the results. The Conclusions are crucial for the publication and therefore should be very clear and not only repeating the results.
  6. Lines 138 and 449: was it written informed consent?

Minor comments:

  1. Discussion: I think it is more clear now. Please state the ref. at the end of the sentences (with exception of lines 305-307, where putting the ref. inside the sentence is understandable).
  2. Figures 1-4, Table 1: lack footnotes with explained abbreviations.
  3. The English language needs major corrections in grammar and style.

- line 371 and following: “attributed to” not “with” (like in line 281).

Author Response

Dear Reviewer,

Thank you for your message and taking the time to revise our paper. We greatly appreciate your assistance and positive feedback in relation to the modified manuscript.

We have now implemented some further changes which have been suggested. These changes have been approved by all the authors of the paper and have been highlighted in the manuscript as requested. Herewith, the revised version.

We have also taken this opportunity to respond to your comments here to ensure the concerns you have raised are addressed appropriately.

Comment 1: In my opinion the title should be modified and it should not include “hospitalized”. As explained by Authors only the CG was hospitalized per se (lines 135-136). SG is the clue of the research and these patients were hospitalized only to perform the study/perform planned tests.

Response 1: Thank you for pointing this out. It is generally accepted practice to refer patients with suspected reactive hypoglycemia to the hospital for more extensive testing and diagnosis. This was the reason why we were investigating our patients in a hospital setting rather than in an outpatient setting.  If you feel there is no need to emphasize the word “hospitalized” in the title, we have all agreed to modify it.  The revised suggested title for our paper is “Metabolic parameters in patients with suspected reactive hypoglycemia.”

Comment 2.: Lines 109-110- there is again inaccuracy, whether the CG was also hospitalized for 3 days? It seems to be different from the information in the lines 135-136.

Response 2: Noted. As we have specified in Materials and Methods (2.1. Participants) all patients were hospitalized for three days (line 123) and these participants were divided into two groups - CG and SG. We have, therefore added additional sentence (line 207) to clarify information about the duration of hospitalization  as it is a standard procedure in our hospital to admit patients for the minimum period of 3 days. 

Comment 3: Lines 96-100: thank you for adding the paragraph. It should contain ref. Moreover, I still think there should be mentioned hypothesised role of lipids in glucose/insulin metabolism (as in lines 387-390) in the introduction, as this is a crucial part of the research.

Response 3: Noted and references have been added.

Additionally, (line 104-112) some further explanation about glucose metabolism and disturbed lipid profile has been added.

Comment 4: I appreciate the changes made in the explaining the study groups. However, Material and Methods section is still mixed: lines 124-136 should go after line 179 as some of the patients were excluded after OGTT, if I understand correctly.

Response 4: Thank you for this comment. However, in our opinion, the reordering and written description of patients in the Anthropometric and laboratory measurements section may cause inconsistency in the text. For this purpose, we have decided to add a separate paragraph (2.3. Final characteristics of participants; line 194) which summarizes the grouping, mentioning exclusion based on OGTT result. In this paragraph, we have also included a reference to Figure 1, which provides a schematic overview of the study.

Comment 5: Lines 171-173: lack ref. As hypoglycaemia and hyperinsulinaemia are the important possible results of the research the reader should be given the ref. for the values adapted in the study.

Response 5: Noted and references have been added. As we emphasized in the discussion, these values vary among studies. In our study, we have relied on the values described in the references, along with the fact that these values are generally accepted and implemented in our hospital. These are the arbitrary values, and we are aware that modifying these values could result in variations in the diagnostic rate.  However, considering the fact that these patients are young, with normal body weight and no comorbidities, we believe that based on current knowledge and physiological fluctuation of glucose and insulin those values seems reasonable.  

Comment 6: I understand that 5-hr OGTT is not a standard OGTT. However, OGTT is a widely known procedure and in my opinion most of the lines 163-170 should be deleted, except the information that: 75g of glucose was used and glucose and insulin levels were measured at baseline and every 60 minutes for a period of five hours.

Response 6: Noted and agreed. We have deleted unnecessary information according to OGTT.

Comment 7: I feel there is lack of possible conclusions from the presented results: lower insulin in RH group suggests adequate insulin response, lower glucose in OGTT in comparison to MMTT suggests the reason for modifying diet in the patients with RH. Therefore, Conclusions in Abstract and the text should be modified, not only revealing the results. The Conclusions are crucial for the publication and therefore should be very clear and not only repeating the results.

Response 7: Thank you, for raising this important issue. As we agree with your opinion, we have decided to make a modification in conclusion (line 36-41; line 467-476).

In addition, we would like to emphasize that the analysis of the dietary model and the implementation of a nutritional intervention with a Low Glycemic Index Diet and the Mediterranean Diet will be the subject of our next study which is currently in progress.

Comment 8: Lines 138 and 449: was it written informed consent?

Response 8: Yes, it was written informed consent. We have now added this information.

The written informed consent and all the questionnaires (written in Polish) that were completed by patients during the study have been approved by the Committee in our hospital.

Minor comments:

Comment 1: Discussion: I think it is more clear now. Please state the ref. at the end of the sentences (with exception of lines 305-307, where putting the ref. inside the sentence is understandable).

Response 1: Noted and all the references have been changed.

Comment 2: Figures 1-4, Table 1: lack footnotes with explained abbreviations.

Response 2: Thank you for this comment. All the footnotes with explained abbreviations have been added.

Comment 3: The English language needs major corrections in grammar and style.

Response 3:  Our article has been now reviewed by a native speaker and necessary corrections have been made.

Comment 4: line 371 and following: “attributed to” not “with” (like in line 281).

Response 4: Corrections have been made.

We trust that the modifications mentioned above are acceptable, but should you wish to add any further comments, we will be grateful for your continuous guidance.

We hope the revised manuscript meets the Journal of Personalized Medicine criteria and requirements and is one step closer to being published.

Thank you for your continued support and interest in our research.

Sincerely,

Marianna Hall
